# Factors Affecting Physical and Mental Fatigue among Female Hospital Nurses: The Korea Nurses’ Health Study

**DOI:** 10.3390/healthcare9020201

**Published:** 2021-02-13

**Authors:** Hee Jung Jang, Oksoo Kim, Sue Kim, Mi Sun Kim, Jung Ah Choi, Bohye Kim, Hyunju Dan, Heeja Jung

**Affiliations:** 1School of Nursing and Research Institute of Nursing Science, Hallym University, Chuncheon 24252, Korea; hjjang@hallym.ac.kr; 2College of Nursing, Ewha Womans University, Seoul 03760, Korea; ohong@ewha.ac.kr (O.K.); bohyekim516@naver.com (B.K.); hidan@hanmail.net (H.D.); 3Ewha Research Institute of Nursing Science, Seoul 03760, Korea; jungah.ch@gmail.com; 4College of Nursing, Yonsei University, 50 Yonsei-ro, Seodaemun-gu, Seoul 03722, Korea; SUEKIM@yuhs.ac; 5Seoul Health Foundation, 31 Maebongsan-ro, Mapo-gu, Seoul 03909, Korea; sunny97kim@seoulhealth.kr; 6College of Nursing, Konyang University, 158 Gwanjeodong-ro, Seo-gu, Daejeon 35365, Korea

**Keywords:** physical fatigue, mental fatigue, female, nurses

## Abstract

Nurses often experience work-related physical and mental fatigue. This study sought to identify the levels of physical and mental fatigue present among Korean female nurses and discern factors influencing their onset. This cross-sectional study analyzed data from the Korea Nurses’ Health Study (KNHS). A total of 14,839 hospital nurses were assessed by hierarchical regression analysis. The mean scores of physical and mental fatigue were 12.57 and 5.79 points, respectively. After adjusting for confounding variables, the work department had a significant influence on both physical and mental fatigue, that is, nurses working in special care units experienced greater degrees of both physical and mental fatigue than those working in general units. Nurse fatigue is an important consideration to monitor to ensure nurses’ continued wellbeing as well as good patient safety levels. Therefore, it is necessary to establish a strategy to mitigate nursing fatigue while considering the characteristics of specific departments. In nursing practice, the introduction of a counseling program and guarantee of rest time that can alleviate the mental and physical fatigue of nurses working in special care units should be considered.

## 1. Introduction

Nurses are required to perform appropriate nursing care and treatment at the frontlines of patient care and, as a result, often experience work-related physical and mental fatigue. According to a prior study, 50.2% of hospital nurses reported work-related chronic and acute fatigue and 84.9% of female nurses have experienced physical and mental fatigue [1,2]. These high proportions of physical and mental fatigue among nurses must be given attention as an important issue. Physical fatigue is caused by physical labor, such as long hours of standing, lifting, or changing the positions of patients, and appears to be a symptom of full-body discomfort and difficulty in tasks requiring strength [3]. In particular, healthcare professionals such as nurses encounter the risks of high exposure to posture-related harm, which can lead to musculoskeletal disorders and become a major factor of physical fatigue [4]. Mental fatigue is caused by work-related emotional stress such as patients’ demands and expectations, which results in lethargy, decreased levels of concentration, or lack of motivation for work [3]. Both physical and mental fatigue negatively affect an individual’s biological, psychological, and cognitive processes [5,6]. High levels of physical and mental fatigue affect nurses’ personal health and health-promoting behavior [7]. Fatigue can also decrease nurses’ work performance and may impair their ability to perpetuate safe behaviors in the workplace [8,9]. Therefore, it can be argued that persistent fatigue is not just a nurse’s personal problem but also an issue that directly affects patient safety and the quality of care. 

According to previous studies, married nurses, those who work in shifts, and those who receive less income display a higher incidence of fatigue [5,10,11]. Nurses with a lower quality of sleep also reported more fatigue, whereas those without depressive symptoms and who had better-perceived health felt less fatigue [2,12]. Level of fatigue varies by work department as well. Emergency room (ER) nurses are more likely to experience high levels of fatigue relative to nurses working in other nursing departments [1,13]. This discrepancy is attributed to the characteristics of these respective special care departments, including the need to manage complicated and life-threatening medical problems, the existence of physically and mentally demanding job requirements, the persistence of increased pressure to perform well, and the use of complex technologies [1,5]. In addition, to the best of our knowledge, none of the previous studies on fatigue by workplace evaluated differences between physical and mental fatigue among nurses according to their work department. As such, there are limitations in identifying differences and in suggesting the relevance between physical and mental fatigue among departments. This study aimed to (1) identify the levels of physical and mental fatigue among Korean nurses and (2) investigate the factors affecting physical and mental fatigue.

## 2. Materials and Methods 

### 2.1. Study Design, Population, and Setting

This study was cross-sectional in design. We analyzed data from the Korea Nurses’ Health Study (KNHS), which is described in detail elsewhere [14]. The KNHS is a large-scale, prospective cohort study of female nurses that investigates the effects of occupational and lifestyle risk factors on Korean women’s health. This study only included female nurses as participants due to the increasing demand to understand women’s health issues, considering the dramatic changes in the fertility rate and social changes arising from the rapid economic growth in Korea. The study population of the KNHS included female hospital nurses of child-bearing aged between 20 and 45 years old. Data were collected online, through the KNHS website, from July 2013 to November 2014. Voluntary participation was encouraged by advertising the study on the KNHS website and social networking sites; furthermore, the research team visited the job training sessions held at the hospitals to encourage personnel to participate in the study. 

A total of 20,613 nurses participated in module 1, the initial baseline survey, and 14,839 nurses who worked in general care departments and special care departments were included in the analysis for this study. Based on the specialty of the job and the traits of the working environment, the emergency room (ER), intensive care unit (ICU), operating room (OR), and post-anesthesia care unit (PACU) were classified as special care departments, while general wards (i.e., internal medicine, general surgery, and obstetrics and gynecology) and outpatient units were classified as general care departments. 

### 2.2. Measurements

The questionnaires of KNHS were similar to those of the Nurses’ Health Study 3 (NHS 3) performed in the United States (US) [14]. A multidisciplinary advisory committee translated the NHS 3 questionnaires, then modified or eliminated some questions to ensure relevance to Korean health research needs and accurately reflect cultural issues. 

Fatigue, the primary variable of interest, was measured using the Chalder Fatigue Scale (CFS). This questionnaire consists of 11 items, where items 1 through 7 measure physical fatigue, while items 8 through 11 measure mental fatigue. Answers are assessed using a four-point Likert scale with zero points indicating less than usual, one point indicating no more than usual, two points indicating more than usual, and three points indicating much more than usual. Total physical fatigue scores can therefore range from zero to 21 points, while total mental fatigue scores range from zero to 12 points, with higher scores indicating more severe levels of fatigue in both cases. The Cronbach’s alpha values of the original study were 0.85 (physical fatigue) and 0.82 (mental fatigue; 14) [15]. The Cronbach’s alpha value of this study was 0.91. 

Stress was measured using the four-item Perceived Stress Scale-4 (PSS-4). Responses were rated on a five-point Likert scale ranging from zero points indicating not at all to four points indicating very often. Possible scores ranged from zero to 16 points and higher total scores suggested the existence of higher levels of perceived stress in respondents. The Cronbach’s alpha of the original study was 0.72 [16], while the Cronbach’s alpha of the current study was 0.52. 

Depressive symptoms were measured via the Patient Health Questionnaire-9 (PHQ-9), a self-reported instrument used to identify the severity of depressive symptoms [17]. The PHQ-9 consists of nine items with a total score that ranges from zero to 27 points. Scores of zero to four points, five to nine points, 10 to 14 points, 15 to 19 points, and 20 points or higher indicate minimal, mild, moderate, moderately severe, and severe depressive symptoms, respectively [17]. The Cronbach’s alpha of the current study was 0.90, which is similar to that of the original paper (0.87). 

We used the Jenkins Sleep Questionnaire to measure sleep disturbance [18]. This questionnaire consists of four items assessed using a six-Likert scale ranging from one point indicating not at all to six points indicating every night. Total scores here can range from four to 24 points and higher scores suggest the respondent is experiencing more severe sleep problems. The Cronbach’s alpha of the original study was 0.79, while that of this study is 0.86 [18]. 

Participants were also asked to rate their health as good, fair, or poor. Separately, to investigate factors affecting nurses’ fatigue, we adjusted for the following general covariates based on a literature review: general characteristics (i.e., age, level of education, marital status, annual income, shiftwork), body mass index (BMI), and sleep and psychological health (sleep problems, perceived health, level of depression, stress). All of these factors were included as potential confounding variables. 

### 2.3. Data Analysis 

The data were analyzed using the SPSS Statistics version 26.0 software program (IBM Corp., Armonk, NY, USA). Descriptive statistics were adopted to examine the frequencies and percentages, while Pearson’s correlation coefficient was used to examine the associations among variables. Factors affecting the levels of physical and mental fatigue among female hospital nurses were analyzed via hierarchical regression analysis. The threshold for statistical significance was *p* < 0.05. 

### 2.4. Ethical Considerations

This study received ethical approval from the Korea Centers for Disease Control and Prevention (2013-03CON-03-P). Anonymity was assured, and informed consent was obtained from the participants online. 

## 3. Results

Table 1 presents the participants’ general characteristics and the differences observed for each variable according to both physical and mental fatigue. Out of the 14,839 nurses, 62.9% were aged 29 years or younger, almost half had completed a four-year university or higher education program (50.9%), and most (70.2%) were single. Approximately half of the participants (42.1%) earned an annual income of lower than $30,000 (USD), while 38.9% earned an annual income of between $30,000 and $39,999 USD. Most of the nurses worked shiftwork (78.3%) and 65.7% had normal BMI values. The majority of the nurses rated their health as either good (41.1%) or fair (48.7%); most of the nurses had no sleep problems (64.9%) and, with regard to depressive symptoms, most said they were experiencing minimal (32.9%) or mild (37.4%) levels of depressive symptoms, while only 3.7% reported severe depressive symptoms. The mean score of stress was 6.61 of 16 points. Considering the work department, 62.0% worked in general units. The mean score of overall fatigue was 18.36 of 33 points, suggesting a moderate level of fatigue was prevalent throughout the study population. More specifically, the mean score of physical fatigue was 12.57 of 21.0 points, whereas that of mental fatigue was 5.79 of 12.0 points. 

The following parameters were statistically different according to mental fatigue (Table 1): age, marital status, annual income, shiftwork, perceived health, sleep problems, level of depression, and stress level. Meanwhile, the following parameters were statistically different according to physical fatigue: age, marital status, level of education, marital status, annual income, shiftwork, BMI, perceived health, sleep problems, level of depression, and stress level. 

Table 2 displays the hierarchical multiple regression results, which were employed to discern which factors affect physical fatigue. As shown in model 3, the work department also significantly influenced physical fatigue (β = 0.027; *p* < 0.001). The adjusted R-square in the final model was 46.7% (F = 996.512; *p* < 0.001), revealing an increase of 42.7% in explanatory power compared with that of model 1. Age, marital status (being married), annual income (≤$2999 USD), shiftwork, BMI, perceived health (both fair and poor), sleep problems, level of depression, and stress level were factors showing a significant association with physical fatigue.

Table 3 describes the results of hierarchical multiple regression, which were used to determine factors affecting mental fatigue (dependent variable). Socio-demographic factors and conduct of shiftwork were included in model 1. Health-related factors were then added in model 2. Finally, model 3 identified that differences between departments were statistically significant concerning mental fatigue. When the work department (independent variable) was added in the final model, the adjusted R-square was 36.5% (F = 651.819; *p* < 0.001), constituting an increase of 35.4% in explanatory power relative to model 1. The independent variable, work department, had significant influence on mental fatigue (β = 0.027; *p* < 0.001). Age, marital status (being married), shiftwork, BMI, perceived health (both fair and poor), sleep problems, level of depressive symptoms, and stress level were factors that had a significant relationship with mental fatigue.

## 4. Discussion

This study aimed to confirm the level of fatigue among Korean female nurses and to identify factors influencing the onset and worsening of physical and mental fatigue. The study results suggest that Korean female nurses show a moderate level of fatigue with average scores of physical fatigue and mental fatigue being 12.57 and 5.79 points, respectively. In particular, nurses working in special care departments showed higher levels of both physical and mental fatigue in comparison with nurses working in general care departments. In a previous study, nurses working in special care departments such as the ER, ICU, and OR reported higher levels of fatigue, which may be because these nurses must care for patients with greater severities of disease or injury and with higher levels of physical dependency [19]. Psychological and physical fatigue are predictors for medical administration errors. Psychological fatigue, above all, is related to incomplete or incorrect documentation of patients, negatively affecting the tasks performed by nurses [20]. Hence, the regular inspection of nurses’ fatigue levels and the development of intervention plans could play important roles in maintaining patient care quality by enhancing nurses’ work performance. 

Hierarchical multiple regression was chosen to attempt to identify the factors influencing the physical and mental fatigue of nurses. Even after adjusting for confounding variables, work department had a significant influence on nurses’ physical and mental fatigue. Some prior studies have reported that the work department is related to nurse fatigue although one study suggested the opposite—that the work department is not a factor influencing the fatigue of nurses [13,19,21]. Such a gap may have occurred due to different classifications of work departments in each study, making direct comparisons between them more difficult. One study suggested that nursing care models may have a bigger impact than the work unit itself on the chronic occupational fatigue of nurses. In particular, nurses in a total patient care model wherein primary nurses perform many functions express a higher level of fatigue than nurses working in a functional nursing care model, where several nurses are given one or two assignments [21]. In this study, as we did not investigate the nursing care type or model of each department, the interpretation of data may be limited in some respects. However, when the results of prior studies are considered, we can infer that the work department is presented as a predictor of fatigue as each department boasts unique degrees of direct nursing care and physically or mentally demanding tasks. A study on physical activity and the level of fatigue of pediatric nurses working in the pediatric ICU and OR found that more than 50% of the physical activity performed by nurses in special care departments are related to their work [22]. Nurses working in the OR and anesthesia recovery room are also required to set up complicated equipment and often have to work while standing up [23]. Such results show that nurses in special care departments have a higher burden of physical activity, which can contribute to physical fatigue. Moreover, pediatric nurses in special care departments have also reported that they experience higher levels of fatigue from perpetuating interpersonal relationships [22]. Hence, the severity of physical work is considered to be significantly related to not only physical fatigue but also to mental fatigue. 

According to a prior study on work-related factors that provoke physical and mental fatigue among nurses, inadequate time available and competing task demands are the factors that most frequently cause mental fatigue. Physical demand tasks, including lifting, pushing, and carrying, were factors that most frequently lead to physical fatigue [24]. Additionally, challenging working conditions, such as the hierarchies among healthcare workers, task orientation, and inflexible divisions of labor, result in occupational health problems among workers [25]. Hence, to reduce both the physical and mental fatigue of nurses, it is important to identify the nature and demands of the work performed within each department. However, in this study, the levels of physical and mental demands or burdens were not investigated and work units such as the ICU, ER, and OR were classified collectively as special departments. Future studies focusing on and further delineating the characteristics and influences of work according to individual work units would be beneficial. 

One of the controlled variables, shiftwork, was also identified as a factor affecting mental and physical fatigue. While one study found that nurses working in shifts experience a high level of fatigue, there is also conflicting evidence concerning the nature of the relationship between nurses’ shiftwork and fatigue. In a study of nurses working in four centers (pediatric, maternity, general, and emergency) at multispecialty hospitals, there was no meaningful relationship established between shiftwork and fatigue [20,26]. Elsewhere, in a literature review of work schedule characteristics and the fatigue profiles of nurses, an insufficient resting period had a more significant relationship with the level of fatigue relative to the number of night or evening shifts [27]. As such, the relationship between shiftwork and fatigue needs to be reviewed further with consideration of the role of rest periods among nurses working in shifts. 

In addition, in this study, the level of depression was found to be an influencing factor for mental and physical fatigue. Few studies have investigated this relationship between the level of depression and fatigue among nurses, despite the higher levels of depressive symptoms exhibited by nurses, compared to other occupations, due to working in shifts [28]. Prior studies on college students revealed that those with moderate or severe fatigue showed significantly higher depressive symptoms and scored higher on a suicide risk measure than those with mild or no fatigue and those with mild fatigue, respectively [29]. Therefore, it is considered that proper management of depressive symptoms among nurses can have a relieving effect on mental and physical fatigue.

Finally, in this study, women in their 20s showed higher mental and physical fatigue than those in their 30s. These results indicate that mental and physical fatigue can be reduced through job adaptation as age and experience increase. However, in this study, nurses’ work experience was not included as a covariate in the analysis; therefore, its relationship with fatigue could not be identified. Thus, it is necessary to understand the relationship between nurses’ work experience and fatigue in future studies. 

Identifying differences in factors that contribute to fatigue in nurses and providing appropriate interventions are important to effectively maintain nurses’ health and reduce nurse turnover, especially in special care departments requiring longer staff training periods. Prior studies have suggested the effectiveness of relaxation exercise using deep inspiration and pursed-lip breathing techniques among ER nurses and the conduct of higher levels of physical activity contributing to a decrease in fatigue scores among pediatric nurses in a special care unit, respectively [22,30]. 

This study is limited due to its cross-sectional nature, which does not allow for an explanation of causal relationships among variables. However, as the KNHS is ongoing, future analyses of collected cohort data are anticipated to potentially shed more light on nurses’ fatigue. Other limitations include not being able to control for direct variables (e.g., time spent standing up and the number of heavy objects lifted) when investigating differences in fatigue, according to work department, and selection issues. As this study only analyzed nurses working in internal medicine, general surgery, obstetrics and gynecology, outpatient units, ER, ICU, OR, and post-anesthesia care units, care is warranted when pursuing interpretations for nurses in other work areas. Finally, despite these limitations, the large sample size and wide participation of nurses nationwide provide significant data for understanding nurses’ fatigue.

## 5. Conclusions

Fatigue in nurses can eventually lead to burn out and start a vicious cycle of deteriorating patient care due to turnover among nurses and increased workload for their colleagues. This potential lack of skilled nurses can be particularly detrimental to the services of special care departments. However, efforts to reduce the fatigue of these nurses have been insufficient. Practical policy measures to assess and mitigate fatigue in nurses are required, not only for nurses’ wellbeing but also for patient safety. Moreover, it should be considered especially important in the ongoing Coronavirus 2019 (COVID-19) pandemic.

## Figures and Tables

**Table 1 healthcare-09-00201-t001:** General characteristics of participants (*N* = 14,839).

Variables			Mental Fatigue		Physical Fatigue	
	N	%	M ± SD	*t* or F	M ± SD	*t* or F
				(*p*)		(*p*)
Total	14,839	100	5.79 ± 2.60		12.57 ± 4.04	
Age				4.479 (0.000)		16.477 (0.000)
≤29	9334	62.9	5.86 ± 2.66		12.99 ± 3.98	
≥30	5505	37.1	5.66 ± 2.50		11.86 ± 4.03	
Level of Education				0.417 (0.677)		3.604 (0.000)
3-year college	7095	49.3	5.80 ± 2.62		12.70 ± 4.04	
4-year university or higher	7744	50.7	5.78 ± 2.59		12.46 ± 4.03	
Marital status				−5.806 (0.000)		−15.287 (0.000)
Married	4420	29.8	5.60 ± 2.50		11.80 ± 4.01	
Single or other status	10,419	70.2	5.86 ± 2.64		12.90 ± 4.00	
Annual income (USD)				4.485 (0.011)		40.900 (0.000)
≤$2999	6250	42.1	5.77 ± 2.70		12.69 ± 4.11	
$3000–3999	5764	38.9	5.85 ± 2.53		12.75 ± 3.93	
≥$4000	2824	19.0	5.68 ± 2.53		11.96 ± 4.04	
Shift work				−11.933 (0.000)		−17.814 (0.000)
No	3224	21.7	5.30 ± 2.57		11.42 ± 4.18	
Yes	11,614	78.3	5.92 ± 2.60		12.89 ± 3.94	
BMI				0.846 (0.429)		78.190 (0.000)
Normal	9707	65.7	5.77 ± 2.61		12.48 ± 4.05	
Underweight	2441	16.5	5.84 ± 2.65		13.44 ± 4.00	
Overweight	2631	17.8	5.80 ± 2.54		12.09 ± 3.92	
Perceived health				1009.34 (0.000)		2266.00 (0.000)
Good	6093	41.1	4.84 ± 2.50		10.49 ± 3.86	
Fair	6681	45.0	6.12 ± 2.34		13.38 ± 3.26	
poor	2065	13.9	7.48 ± 2.60		16.11 ± 3.34	
Sleep problem				−43.162 (0.000)		−60.598 (0.000)
No	9626	64.9	5.14 ± 2.44		11.28 ± 3.79	
Yes	5213	35.1	6.97 ± 2.48		14.95 ± 3.35	
Level of Depression				1517.243 (0.000)		1848.911 (0.000)
Minimal	4872	32.9	4.14 ± 2.25		9.72±3.73	
Mild	5540	37.4	5.86 ± 2.15		12.81 ± 3.08	
Moderate	2598	17.5	6.92 ± 2.13		14.59 ± 2.98	
Moderate severe	1266	8.5	7.94 ± 2.16		16.12 ± 3.06	
Severe	547	3.7	9.31 ± 2.39		17.68 ± 3.20	
Stress level				728.520 (0.000)		620.881 (0.000)
1st quartile	4582	30.9	4.62 ± 2.35		10.95 ± 3.79	
2nd quartile	4385	29.6	5.78 ± 2.24		12.65 ± 3.41	
3rd quartile	3416	23.0	6.21 ± 2.68		12.88 ± 4.38	
4th quartile	2456	16.5	7.37 ± 2.54		15.02 ± 3.65	
Department				−3.227 (0.001)		−3.759 (0.000)
General	9198	62.0	5.73 ± 2.62		12.47 ± 4.09	
Special	5641	38.0	5.87 ± 2.58		12.73 ± 3.95	

Note: BMI = body mass index; M = mean; SD = standard deviation.

**Table 2 healthcare-09-00201-t002:** Hierarchical multiple regression analysis predicting physical fatigue (*N* = 14,839).

Variables		Model 1		Model 2		Model 3	
		β	*t*	β	*t*	β	*t*
Age		−0.146 ***	−12.049	−0.065 ***	−7.085	−0.065 ***	−7.020
Level of Education (4-year university or higher = 0)							
	3-year college	−0.008	−0.937	−0.006	−0.874	−0.005	−0.824
Marital status (Single or others = 0)							
	Married	−0.014	−1.287	0.066 ***	8.190	0.066 ***	8.247
Annual income (USD) (≥$4000 = 0)							
	$3000–3999	0.003	0.242	−0.003	−0.359	−0.003	−0.321
	≤$2999	−0.037 **	−2.739	−0.033 **	−3.199	−0.030 **	−2.953
Shift work (No = 0)							
	Yes	0.102 ***	11.843	0.025 ***	3.845	0.024 ***	3.764
BMI				−0.095 ***	−15.295	−0.095 ***	−15.334
Perceived health (Good = 0)							
	Fair			0.226 ***	33.546	0.225 ***	33.496
	Poor			0.267 ***	37.664	0.267 ***	37.686
Sleep problem				0.157 ***	20.524	0.156 ***	20.463
Level of Depression				0.380 ***	45.484	0.380 ***	45.538
Stress level				0.040 ***	5.665	0.040 ***	5.765
Department (General = 0)							
	Special					0.027 ***	4.430
*R* ^2^		0.041		0.467		0.468	
Adjusted *R*^2^		0.040		0.466		0.467	
*F*		104.760 ***		1076.560 ***		996.512 ***	
Δ*R*^2^				0.426		0.001	

Note: BMI = body mass index; *** *p* < 0.001, ** *p* < 0.01.

**Table 3 healthcare-09-00201-t003:** Hierarchical multiple regression analysis predicting mental fatigue (*N* = 14,839).

Variables		Model 1		Model 2		Model 3	
		β	*t*	β	*t*	β	*t*
Age		−0.020	−1.603	0.060 ***	6.012	0.061 ***	6.078
Level of Education (4-year university or higher = 0)							
	3-year college	−0.008	−0.935	−0.007	−0.975	−0.006	−0.929
Marital status (Single or others = 0)							
	Married	−0.014	−1.280	0.052 ***	5.931	0.052 ***	5.982
Annual income (USD) (≥$4000 = 0)							
	$3000–3999	0.009	0.725	−0.002	−0.216	−0.002	−0.181
	≤$2999	−0.010	−0.713	−0.021	−1.845	−0.018	−1.621
Shift work (No = 0)							
	Yes	0.089 ***	10.162	0.023 **	3.235	0.022 **	3.160
BMI				−0.021 **	−3.102	−0.022 **	−3.131
Perceived health (Good = 0)							
	Fair			0.099 ***	13.519	0.099 ***	13.463
	Poor			0.118 ***	15.293	0.118 ***	15.299
Sleep problem				0.089 ***	10.722	0.089 ***	10.661
Level of Depression				0.423 ***	46.339	0.423 ***	46.386
Stress level				0.133 ***	17.414	0.134 ***	17.508
Department (General = 0)							
	Special					0.027 ***	4.043
*R* ^2^		0.011		0.364		0.365	
Adjusted *R*^2^		0.010		0.364		0.364	
*F*		26.282 ***		704.042 ***		651.819 ***	
Δ*R*^2^				0.354		0.001	

Note: BMI = body mass index; *** *p* < 0.001, ** *p* < 0.01.

## Data Availability

The datasets generated and analyzed during the current study are not publicly available because this government data needs time for data clearing and establishment of guidelines. The Korea Centers for Disease Control and Prevention is planning on opening this data to the public in the future.

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
