# Peer review of "Factors Affecting Physical and Mental Fatigue among Female Hospital Nurses: The Korea Nurses’ Health Study"

_healthcare, 2021, doi:10.3390/healthcare9020201_

Round 1
Reviewer 1 Report
The present manuscript entitled "Factors Affecting Physical and Mental Fatigue Among Female Hospital Nurses: The Korea Nurses’ Health Study "is an interesting and relevant study, but it has certain limitations that are detailed below:
- In the Material and Methods section, no mention is made of ethical considerations in a subsection, it comes at the end of the document.
It would be convenient to explain how the data was collected and used. Are they anonymous?
- In the tables presented there is no footnote in which the acronyms are developed. This makes it difficult to read and understand its content.
- There is no conclusions section with future lines.
- Throughout the manuscript, mention is made of female nurses, why has only the female sex been studied? There is no mention in the manuscript of that gender choice.
Author Response
♦ Reviewer # 1
The present manuscript entitled "Factors Affecting Physical and Mental Fatigue Among Female Hospital Nurses: The Korea Nurses’ Health Study "is an interesting and relevant study, but it has certain limitations that are detailed below.
- In the Material and Methods section, no mention is made of ethical considerations in a subsection, it comes at the end of the document.
Authors added ethical considerations as follows.
“This study received ethical approval from the Korea Centers for Disease Control and Prevention (2013-03CON-03-P).” (Ethical Consideration, page 3, line 142-144).
- It would be convenient to explain how the data was collected and used. Are they anonymous?
Authors added following sentences in the Study Design, Population, and Setting section. Also the content of guaranteeing the anonymity of survey participants has also been added to the ethical consideration section.
“Data were collected online, through the KNHS website, from July 2013 to November 2014. Voluntary participation was encouraged by advertising the study on the KNHS website and social networking sites; furthermore, the research team visited the job training sessions held at the hospitals to encourage personnel to participate in the study.” (Study Design, Population, and Setting, page 2, line 82-86).
“Anonymity was assured, and informed consent was obtained from the participants online.” (Ethical Consideration, page 3, line 144-146).
- In the tables presented there is no footnote in which the acronyms are developed. This makes it difficult to read and understand its content.
We added footnotes in each table. Thank you for your comment.
- 4. There is no conclusions section with future lines.
Conclusion section was added as follows. ” (Conclusion, page 9, line 299-307).
“ 5. Conclusions
Fatigue in nurses can eventually lead to burn out and start a vicious cycle of deteriorating patient care due to turnover among nurses and increased workload for their colleagues. This potential lack of skilled nurses can be particularly detrimental to the services of special care departments. However, efforts to reduce the fatigue of these nurses have been insufficient. Practical policy measures to assess and mitigate fatigue in nurses are required, not only for nurses’ wellbeing but also for patient safety. Moreover, it should be considered especially important in the ongoing Coronavirus 2019 (COVID-19) pandemic.”
- Throughout the manuscript, mention is made of female nurses, why has only the female sex been studied? There is no mention in the manuscript of that gender choice.
In response to reviewer’s comment, authors added following sentence.
“This study only included female nurses as participants due to the increasing demand to understand women's health issues, considering the dramatic changes in the fertility rate and social changes arising from the rapid economic growth in Korea.” (Study Design, Population, and Setting, page 2, line 78-81).

Reviewer 2 Report
The authors present a correct and interesting study on the incidence of physical and mental fatigue in a sample of female nurses, analyzing the factors most related to these risks. There are a few areas this manuscript can be improved.
- Introduction. I see it necessary to briefly comment on the presence of tiring and painful possitions, which is one of the most significant factors in the onset of physical fatigue (see page 45; https://www.eurofound.europa.eu/sites/default/files/ef_publication/field_ef_document/ef1634en.pdf) and use the term Musculoskeletal Disorders.
- I consider it necessary to deepen the relationship between the level of depression and mental and physical fatigue., which is not covered by the analysis of the results.
- Was the previous experience or age of the sample evaluated and its relationship to the incidence of fatigue assessed? perhaps, it could be commented on in the conclusions.
- Is the current social-health situation caused by the progress of COVID 19 considered to have aggravated the situation? should be mentioned as future Address
- The conclusions section can also be condensed by removing some repeated result description but just focus on recommendation and policy implications.
Author Response
♦ Reviewer # 2
The authors present a correct and interesting study on the incidence of physical and mental fatigue in a sample of female nurses, analyzing the factors most related to these risks. There are a few areas this manuscript can be improved.
- Introduction. I see it necessary to briefly comment on the presence of tiring and painful possitions, which is one of the most significant factors in the onset of physical fatigue (see page 45; https://www.eurofound.europa.eu/sites/default/files/ef_publication/field_ef_document/ef1634en.pdf) and use the term Musculoskeletal Disorders.
According to reviewer’s comment, authors added following sentence and reviewer’s recommended reference in the Introduction section.
“In particular, healthcare professionals such as nurses encounter the risks of high exposure to posture-related harm, which can lead to musculoskeletal disorders and become a major factor of physical fatigue [4].” (Introduction, page 1-2, line 45-48).
- I consider it necessary to deepen the relationship between the level of depression and mental and physical fatigue., which is not covered by the analysis of the results.
According to reviewer’s comment, authors added following sentences in discussion section
“In addition, in this study, the level of depression was found to be an influencing factor for mental and physical fatigue. Few studies have investigated this relationship between the level of depression and fatigue among nurses, despite the higher levels of depressive symptoms exhibited by nurses, compared to other occupations, due to working in shifts [28]. Prior studies on college students revealed that those with moderate or severe fatigue showed significantly higher depressive symptoms and scored higher on a suicide risk measure than those with mild or no fatigue and those with mild fatigue, respectively [29]. Therefore, it is considered that proper management of depressive symptoms among nurses can have a relieving effect on mental and physical fatigue.” (Discussion, page 8-9, line 264-272).
- Was the previous experience or age of the sample evaluated and its relationship to the incidence of fatigue assessed? perhaps, it could be commented on in the conclusions.
According to the reviewer's comment, the following sentences was added to the main text, and authors decided that it was more appropriate to include it in the discussion section rather than the conclusion section.
“Finally, in this study, women in their 20s showed higher mental and physical fatigue than those in their 30s. These results indicate that mental and physical fatigue can be reduced through job adaptation as age and experience increase. However, in this study, nurses’ work experience was not included as a covariate in the analysis, and therefore, its relationship with fatigue could not be identified. Thus, it is necessary to understand the relationship between nurses’ work experience and fatigue in future studies.” (Discussion, page 9, line 273-279).
- Is the current social-health situation caused by the progress of COVID 19 considered to have aggravated the situation? should be mentioned as future Address.
According to reviewer’s comment, authors added following sentence in conclusion section.
“Moreover, it should be considered especially important in the ongoing Coronavirus 2019 (COVID-19) pandemic.” (Conclusion, page 9, line 305-307).
- The conclusions section can also be condensed by removing some repeated result description but just focus on recommendation and policy implications.
Authors added conclusion section. Thank you for your comment.
- Conclusions
“Fatigue in nurses can eventually lead to burn out and start a vicious cycle of deteriorating patient care due to turnover among nurses and increased workload for their colleagues. This potential lack of skilled nurses can be particularly detrimental to the services of special care departments. However, efforts to reduce the fatigue of these nurses have been insufficient. Practical policy measures to assess and mitigate fatigue in nurses are required, not only for nurses’ wellbeing but also for patient safety. Moreover, it should be considered especially important in the ongoing Coronavirus 2019 (COVID-19) pandemic.”(Conclusion, page 9, line 299-307).
Reviewer 3 Report
The article discusses the factors affecting physical and mental fatigue among female hospital nurses in Korea. The article is generally written well. However, I feel that there are things missing that could strengthen the paper further:
- The line numbering has been removed in this version that I reviewed. Please keep the line numbering.
- Re: introduction, last line, “…which results in a sense of boredom”. Mental fatigue is more than “boredom”. Please revise this statement.
- Re: page 2, paragraph 1 – it also affects patient's quality of care. Please add "and it also affects patient's quality of care" after “patient safety”.
- Page 2, paragraph 3, re: “This study sought”…usually paragraphs are 3 to 5 sentences. Please either merge this with the previous paragraph, or expand upon it and add a few more sentences.
- Page 2, Section 2.2: do you have a citation to corroborate this connection to the U.S. study?
- Table 1: please remove the (p) from under the M+- SD
- Table 1: please add the (p) under T or F under mental fatigue
- Table 1, re: “Single or others” – I am not sure what “others” is supposed to mean. Please revise e.g. “single or other status”
- Table 1, re: “1quatile”, 2quatile, 3quatile, 4quatile…. These should be “quartile”.
- Page 7, paragraph 2: I really think this section could benefit with the work researchers who discuss job demand, and sufficiency of time to do the work. It may be worthwhile to cite the work of Syed, I., Daly, T., Armstrong, P., Lowndes, R., Chadoin, M., Naidoo, V. (2016). How Do Work Hierarchies and Strict Divisions of Labour Impact Care Workers' Experiences of Health and Safety? Case Studies of Long Term Care in Toronto. Journal of Nursing Home Research Sciences. 2:41-49. PMCID: PMC5218838 Available from: http://dx.doi.org/10.14283/jnhrs.2016.6 . Also useful could be: Samra, J., Gilbert, M., Shain, M., & Bilsker, D. 2012. Psychosocial risk factors. Centre for Applied Research in Mental Health and Addiction (CARM).
If you decide to incorporate these revisions, please upload a manuscript that contains tracked changes or other method to highlight revisions. Thank you for the opportunity to review this work.
Author Response
♦ Reviewer # 3
The article discusses the factors affecting physical and mental fatigue among female hospital nurses in Korea. The article is generally written well. However, I feel that there are things missing that could strengthen the paper further:
- The line numbering has been removed in this version that I reviewed. Please keep the line numbering.
Authors added line numbers.
- Re: introduction, last line, “…which results in a sense of boredom”. Mental fatigue is more than “boredom”. Please revise this statement.
Authors revised the statement as follows according to reviewer’s comment.
" Mental fatigue is caused by work-related emotional stress such as patients’ demands and expectations, which results in lethargy, decreased levels of concentration, or lack of motivation for work [3].”(Introduction, page 2, line 49).
- Re: page 2, paragraph 1 – it also affects patient's quality of care. Please add "and it also affects patient's quality of care" after “patient safety”
The sentence was revised as follows by the reviewer’s comment.
“Therefore, it can be argued that persistent fatigue is not just a nurse’s personal problem but also an issue that directly impacts patient safety and the quality of care.”(Introduction, page 2, line 56).
- Page 2, paragraph 3, re: “This study sought”…usually paragraphs are 3 to 5 sentences. Please either merge this with the previous paragraph, or expand upon it and add a few more sentences.
Authors merged the sentence with previous paragraph.
“In addition, to the best of our knowledge, ….physical and mental fatigue among departments. This study aimed to (1) identify the levels of physical and mental fatigue among Korean nurses and (2) investigate the factors affecting physical and mental fatigue.”
- Page 2, Section 2.2: do you have a citation to corroborate this connection to the U.S. study?
Jorge E Chavarro is the principle investigator of US Nurses’ Health Study 3 and he participated in following design paper of KNHS as a co-author.
13. Kim, O.; Ahn, Y.; Lee, H.Y.; Jang, H.J.; Kim, S.; Eun Lee, J., Jung, H., Cho, E.; Lim, J-Y.; Kim, M-J.;…Park, H.Y. The Korea nurses’ health study: a prospective cohort study. J Womens Health (Larchmt) 2017, 26, 892–899. doi:10.1089/jwh.2016.6048. |
We added a reference as follows.
“The questionnaires of KNHS were similar to those of the Nurses’ Health Study 3 (NHS 3) performed in the United States (US) [13].”
- Table 1: please remove the (p) from under the M+- SD
Table 1: please add the (p) under T or F under mental fatigue
Table 1, re: “Single or others” – I am not sure what “others” is supposed to mean. Please revise e.g. “single or other status”
Table 1, re: “1quatile”, 2quatile, 3quatile, 4quatile…. These should be “quartile”
Authors revised Table 1 according to reviewer’s comments.
- Page 7, paragraph 2: I really think this section could benefit with the work researchers who discuss job demand, and sufficiency of time to do the work. It may be worthwhile to cite the work of Syed, I., Daly, T., Armstrong, P., Lowndes, R., Chadoin, M., Naidoo, V. (2016). How Do Work Hierarchies and Strict Divisions of Labour Impact Care Workers' Experiences of Health and Safety? Case Studies of Long Term Care in Toronto. Journal of Nursing Home Research Sciences. 2:41-49. PMCID: PMC5218838 Available from: http://dx.doi.org/10.14283/jnhrs.2016.6 . Also useful could be: Samra, J., Gilbert, M., Shain, M., & Bilsker, D. 2012. Psychosocial risk factors. Centre for Applied Research in Mental Health and Addiction (CARM).
Authors added following sentence in the discussion section as reviewer’s comment and include a reference.
“Additionally, challenging working conditions, such as the hierarchies among healthcare workers, task orientation, and inflexible divisions of labor, result in occupational health problems among workers [25].”(Discussion, page 8, line 243-246).

Round 2
Reviewer 3 Report
The article discusses the factors affecting physical and mental fatigue among female hospital nurses in Korea. The article is generally written and articulated well. The authors have followed all of my recommendations for revision and addressed all queries/questions. The manuscript has improved significantly. I recommend that the manuscript should be accepted for publication with the following minor request:
- In Table 1, under “Mental fatigue” for “Sleep problem" and “Yes”, it is written as “(,000)” . This comma should be a period, like this “(.000)”
Congratulations, and thank you for the opportunity to review your work.